# Mucin-Protected Caco-2 Assay to Study Drug Permeation in the Presence of Complex Biorelevant Media

**DOI:** 10.3390/pharmaceutics14040699

**Published:** 2022-03-24

**Authors:** Dong Ye, Álvaro López Mármol, Verena Lenz, Patricia Muschong, Anita Wilhelm-Alkubaisi, Manuel Weinheimer, Mirko Koziolek, Kerstin A. Sauer, Loic Laplanche, Mario Mezler

**Affiliations:** 1Drug Metabolism and Pharmacokinetics—Bioanalytical Research, AbbVie Deutschland GmbH & Co. KG, 67061 Ludwigshafen, Germany; dong.ye@abbvie.com (D.Y.); patricia.muschong@abbvie.com (P.M.); anita.wilhelm-alkubaisi@abbvie.com (A.W.-A.); manuel.weinheimer@abbvie.com (M.W.); loic.laplanche@abbvie.com (L.L.); 2NCE Formulation Sciences, AbbVie Deutschland GmbH & Co. KG, 67061 Ludwigshafen, Germany; alvaro.lopezmarmol@abbvie.com (Á.L.M.); verena.lenz@abbvie.com (V.L.); mirko.koziolek@abbvie.com (M.K.); kerstin.sauer@abbvie.com (K.A.S.)

**Keywords:** permeation, Caco-2 cells, drug absorption, biorelevant media, bioavailability, oral drug delivery

## Abstract

The poor solubility and permeability of compounds beyond Lipinski’s Rule of Five (bRo5) are major challenges for cell-based permeability assays. Due to their incompatibility with gastrointestinal components in biorelevant media, the exploration of important questions addressing food effects is limited. Thus, we established a robust mucin-protected Caco-2 assay to allow the assessment of drug permeation in complex biorelevant media. To do that, the assay conditions were first optimized with dependence of the concentration of porcine mucin added to the cells. Mucin-specific effects on drug permeability were evaluated by analyzing cell permeability values for 15 reference drugs (BCS class I–IV). Secondly, a sigmoidal relationship between mucin-dependent permeability and fraction absorbed in human (*f*_a_) was established. A case study with venetoclax (BCS class IV) was performed to investigate the impact of medium complexity and the prandial state on drug permeation. Luminal fluids obtained from the tiny-TIM system showed a higher solubilization capacity for venetoclax, and a better read-out for the drug permeability, as compared to FaSSIF or FeSSIF media. In conclusion, the mucin-protected Caco-2 assay combined with biorelevant media improves the mechanistic understanding of drug permeation and addresses complex biopharmaceutical questions, such as food effects on oral drug absorption.

## 1. Introduction

In 1997, Christopher Lipinski proposed the “Rule of Five,” which established a link between drug physicochemical properties and their oral bioavailability through a simple rule of thumb for estimating the druggability of new drugs [1]. Almost 25 years later, the number of drugs violating these rules (beyond the Rule of Five or bRo5) that enter the market has increased. These drugs are mostly characterized by poor aqueous solubility and poor permeability, thus classifying them as Class Four of the Biopharmaceutical Classification System (BCS). These properties can pose serious challenges to drug product development [2], and, therefore, the successful selection of drug candidates with respect to druggability requires a precise prediction in terms of drug solubility and permeability.

Drug permeability can be assessed by various in vitro, ex vivo and in vivo methods. The in vitro evaluation of permeability offers the possibility to apply relatively simple and cost-effective models. Here, it can be distinguished from cell-free permeation assays that are based on artificial membranes or an organic phase, such as octanol [3], as well as cell-based assays based on specific cell lines, such as MDCK or Caco-2 cells [4]. In general, cell-based assays (e.g., Caco-2 cells) are considered to have higher physiological relevance, since they do not only consider drug absorption by means of passive diffusion (as in cell-free permeation assays), but also active transport processes that involve uptake and efflux transporters [4].

The standard approach to assess drug permeability across Caco-2 monolayers is based on the apical to basolateral flux of the drug measured within physiological buffers, such as Hank’s Balanced Salt Solution (HBSS) [5]. However, for most drugs in industry pipelines, the aqueous solubility is poor, and their advancement to clinical development is the outcome of an improved understanding of the approaches for promoting their oral exposure [6]. Consequently, simple aqueous buffers may not be sufficient for application in Caco-2 cell experiments, as the concentration of the drug in solution is very low due to the lack of solubilizing components. Addition of biorelevant media addresses these concerns through its higher solubilization capacity and customary use in solubility assessments [7]. 

Various biorelevant media of differing complexities are often used for solubility and dissolution experiments. The standard media applied by most academic and industry groups are based on simulated intestinal fluid (SIF) powder and include fasted state simulated intestinal fluid (FaSSIF) and fed state simulated intestinal fluid (FeSSIF) [8,9]. Recent studies have demonstrated that solubility data in FaSSIF and FeSSIF can differ significantly [10]. Moreover, permeation experiments with diffusion cells performed in the same study also suggest that the activity (i.e., effective concentration of species in solution under non-ideal conditions) of the drugs tested differs as a function of the applied biorelevant media [10]. The choice of medium is therefore expected to affect the assessment of drug permeability. This additional consideration of biorelevant media, historically neglected and substituted with preclinical assessment of drug permeability, shall be considered in this study.

One major limitation to the application of biorelevant media in Caco-2 cells is the inability of this cell line to form a protective layer, i.e., mucus [11]. Lack of mucus can cause the loss of cell monolayer integrity in the presence of higher bile salt concentration or digestive enzymes [12]. Although mucus-protected cell models, such as co-cultured Caco-2 with mucus-secreting HT29-MTX cells, have been proposed [13], abnormally high paracellular transport was also reported, which may hinder their broader application [14]. Moreover, the application of complex biorelevant media simulating the luminal fluid composition in the small intestine is typically not possible in cell-based permeation assays. To overcome this limitation, some authors propose protecting Caco-2 cells with mucins derived from animals (e.g., porcine type III mucin). For instance, Wuyts and colleagues illustrated the protective effect of mucin at a concentration of 50 mg/mL on the Caco-2 cell monolayer in the presence of fasted state human intestinal fluid (FaHIF) over 2 h [15]. However, the performance of such models in the presence of fed state intestinal fluid is not yet known, due to the limited accessibility of prandial-state fluidic conditions in humans. Therefore, in vitro systems mimicking prandial state-dependent absorption becomes important for predicting human effects.

It was the primary aim of this work to assess the in vitro permeability in biorelevant media by using mucus-protected Caco-2 cells. Firstly, how standard media (FaSSIF/FeSSIF) and intestinal fluids (generated through in vitro experiments with tiny-TIM, a complex tool to simulate the gastrointestinal, or GI, conditions in the human stomach and small intestine [16]) interact with cell layer integrity was evaluated. Secondly, the effect of mucin on drug permeation was assessed by comparing permeation data from mucin-protected Caco-2 assays with human fraction absorbed (*f_a_*) values reported for various reference drugs. Finally, this optimized protocol was applied to venetoclax (ABT-199) as a case study on permeation for a poorly water soluble and poorly permeable bRo5 drug (BCS class IV).

## 2. Material and Methods

### 2.1. Chemicals and Reagents

Fifteen reference drugs, including caffeine, piroxicam, theophylline, naproxen, indomethacin, metoprolol, propranolol, carvedilol, indinavir, furosemide, ranitidine, sulfasalazine, atenolol, doxorubicin and cromolyn, were purchased from Sigma-Aldrich (Munich, Germany). ABT-199 was provided by the Global Compound Logistics and Operations Department at AbbVie Inc. (North Chicago, IL, USA). 

Cell culture reagents, including Dulbecco’s Modified Eagle’s Medium (DMEM), fetal bovine serum (FBS), sodium pyruvate, L-Glutamine and non-essential amino acids (NEAA), were all purchased from Life Technologies (St. Leon-Rot, Germany). HBSS (plus CaCl_2_, MgCl_2_), bovine serum albumin (BSA) and HEPES solution were obtained from Gibco, Thermo Fisher Scientific (Dreieich, Germany). Porcine mucin (type III) was purchased from Sigma-Aldrich (Taufkirchen, Germany). Acetonitrile was purchased from Biosolve (The Netherlands). 

SIF powder was purchased from biorelevant.com (London, UK). For the generation of the TIM-media, acetic acid (96%), calcium chloride dihydrate, disodium hydrogen phosphate, hydrochloric acid 1 M, potassium chloride, sodium chloride, sodium hydrogen carbonate, sodium dihydrogen phosphate monohydrate and sodium hydroxide solution 1 M were purchased from Merck KGaA (Darmstadt, Germany). Sodium acetate trihydrate and sodium hydroxide (pellets) were obtained from Honeywell International Inc. (Seelze, Germany). Potassium chloride was purchased from Fluka Analytical (Munich, Germany). (Hydroxypropyl)methyl cellulose (HPMC) was purchased from Colorcon GmbH (Idstein, Germany). Additionally, α-amylase from Bacillus sp.-Type II-A, lipase from Rhizopus oryzae, pancreatin from porcine pancreas (4× USP specifications), pepsin from porcine gastric mucosa, porcine bile extract, sodium chloride, sodium citrate tribasic dihydrate and trypsin from porcine pancreas were obtained from Sigma-Aldrich (Schnelldorf, Germany). Porcine bile was provided by The TIM Company (Druten, The Netherlands). 

### 2.2. Cell Culture

Caco-2 cells were obtained from acCELLerate GmbH (Hamburg, Germany) and were cultured according to the vendor’s instructions. 

In detail, the cell cryogenic vials were thawed in a water bath at 37 °C for 2 min. Immediately, the cells were transferred into one 15 mL tube pre-filled with warm DMEM, which was supplemented with 10% FBS, 1% NEAA, 1% sodium pyruvate and 1% L-Glutamine. The cells were subsequently centrifugated for 4 min at 80× *g*, and the supernatant was cautiously removed. The cell pellet was then gently resuspended in 10 mL of culture medium. After equilibration at room temperature for 30 min, Caco-2 cells were prepared at a density of 400,000 cells/mL and seeded in a volume of 100 µL per well into the apical compartments of 0.4 µm Millicell^®^ 96 cell culture insert plates (Merck Millipore, Germany). The basolateral compartment from the Millicell^®^ feeder tray was filled with 28 mL cell culture medium. The cells were cultivated at 37 °C and 5% CO_2_ in a humidified incubator for 7–8 days, in contrast to the conventional 21-day culture procedure for Caco-2. For cell medium change, fresh medium was exchanged in the basolateral side only on the third day. One day before the assay, culture medium was changed from the apical side, and the basolateral feeder tray was replaced with a 96-well receiver plate (Millicell^®^ 96 Receiver Tray), which was pre-filled with 200 µL of cell medium per well. Depending on the assay time-schedule, the Caco-2 monolayer was replenished with fresh medium one day before the experiment to prevent cell starvation.

### 2.3. Cell Compatibility with Biorelevant Media and Evaluation of Cell Protection by Mucin

The compatibility with different biorelevant media was assessed by measuring the permeability of a paracellular integrity marker, Lucifer Yellow (LuY), in the Caco-2 monolayer.

In general, HBSS at pH 6.5, FaSSIF, FeSSIF, TIM-fasted and TIM-fed media were spiked with LuY for a working concentration at 80 µM. Subsequently, 100 µL of the media was independently added to the apical compartments for Caco-2 treatment, and the basolateral chamber was filled with 200 µL of HBSS pH 7.4 + 1% *w*/*v* of BSA buffer instead. 

For the experiments with mucin, type III porcine mucin powder was first dissolved in HBSS pH 6.5 buffer at two concentrations of 50 and 100 mg/mL, then mixed thoroughly at 37 °C before 15 µL of mucin was applied to the apical side of the Caco-2 monolayer. The mucin–Caco-2 plate was vortexed for 1 min on an orbital shaker at 300 rpm and incubated at 37 °C for 10 min. Afterwards, LuY plus the respective biorelevant media were added on top of the mucin layer, and the HBSS + BSA was filled into the basolateral side. 

At the end of the studies, with or without mucin, 20 µL of LuY, diluted in 180 µL of HBSS, was sampled from the basolateral side of the transwell plate every 1 h for 4 h. The fluorescence intensity was measured with 485/535 nm (excitation/emission) using a plate reader, Tecan GENios(Crailsheim, Germany). The baseline fluorescence intensity was determined for each LuY-spiked biorelevant medium stock. The blank biorelevant media were measured and subtracted from the measured fluorescence values. The % LuY permeation was calculated to compare the Caco-2 integrity changes by using the following equation:(1)$$\%\;\mathrm{initial}\;\mathrm{LuY}\;\mathrm{intensity}=\frac{\;\left(\mathrm{Basolateral}\;\mathrm{sample}\;\mathrm{fluorescence}-\mathrm{Blank}\;\mathrm{control}\right)}{\left(\mathrm{Stock}\;\mathrm{fluorescence}-\mathrm{Blank}\;\mathrm{control}\right)}\times 100\;$$

### 2.4. Permeability of Reference Drugs in Biorelevant Media at Different Mucin Levels

In order to investigate the role of mucin on permeability, we conducted further permeability studies using 15 reference drugs that were also used in a similar study by Wuyts et al. [15]. As shown in Table 1, these drugs consisted of different BCS Classes (I–IV), featured by their diverse permeability values (low to high) and human intestinal drug absorption values (*f*_a_ = 1–100%).

In terms of their physicochemical properties, 13 of the 15 drugs were lipophilic (logP 0.5–3.9) and poorly soluble (≤1.1 mg/mL in water). Only caffeine and theophylline had a logP < 0 and aqueous solubility of 11 and 22.9 mg/mL, respectively. Moreover, 8 of the 15 drugs exhibited a positive electrostatic charge at pH 7.4, while the rest of the drugs were either neutrally (0) or negatively charged (−1 to −2). The majority of the 15 drugs are known to be absorbed across the intestinal epithelia via the transcellular pathway. 

Drug permeability was determined in a mucin-protected Caco-2 model using one 96-format Biomek^®^ FXp liquid-handling system from Beckman Coulter (Brea, California, USA). The detailed assay procedure is shown in Figure 1. 

For each of the 15 reference drugs, a 10 mM stock solution in DMSO was prepared and further diluted to 10 µM by using the aforementioned biorelevant media, as well as HBSS pH 6.5 as the control. The reference drugs were mixed with 80 µM of LuY as the internal control for the monolayer integrity. 

For the Caco-2 transport assay, the culture medium was completely removed from the transwell plate, and the monolayer was rinsed with fresh HBSS pH 7.4. Subsequently, 15 µL of 100 mg/mL porcine mucin was added manually to the apical side of the 96-transwell plate, while fresh HBSS buffer was added to the basolateral compartments. The plate was agitated on an orbital shaker at 300 rpm for 1 min, prior to a short incubation in a humidified incubator at 37 °C for 10 min. After the equilibrium, the monolayers were transferred to a fresh receiver plate. The respective biorelevant medium for the study (either HBSS pH6.5, FaSSIF, FeSSIF, TIM-Fasted or TIM-Fed medium systems), together with the reference drugs, were transferred accordingly onto the apical side of the Caco-2 monolayers. Immediately at t = 0 h, the apical donor samples were collected by the robotic liquid handing system. The basolateral chambers were subsequently filled with HBSS pH 7.4 with a supplement of 1% BSA (*w*/*v*). The transwell plate was incubated at 37 °C in a humidified incubator for 2 h.

At the end of the assay, the apical and basolateral samples were collected; the BSA was precipitated with 100% ethanol containing 0.025 µM of Carbutamide as the internal standard for HPLC–MS/MS analysis, according to the following volume ratios: basolateral side: 25 µL of sample + 100 µL of 100% ethanol; apical side: 5 µL of sample diluted with 20 µL of HBSS pH 7.4 plus 1% BSA + 100 µL of 100% ethanol. The quenched samples were kept frozen for a minimum of 12 h, before they were thawed and centrifuged at 3000× *g* for 30 min at 4 °C to collect the BSA-free supernatants for drug analysis. 

To determine the drug apparent permeability (*P_app_*), the apical and basolateral samples were measured for peak area values in LC–MS/MS and used for calculation according to the following equation:(2)$$P_{app}=\frac{\Delta Q}{\Delta t\cdot A\cdot \left(\frac{c_{1}+c_{0}}{2}\right)}$$
where ΔQ is the amount of drug permeated through the monolayer as determined by the concentration of drug in the receiver well at the end of the experiment, Δt is the incubation time, A is the filter surface area (0.11 cm^2^), C_1_ is the measured concentration in the donor well at the end of the experiment and C_0_ is the initial drug concentration at t = 0. Permeation is expressed as the apparent permeability (*P_app_*) with a unit of 10^−6^ cm/s. 

### 2.5. Solubility Determination of ABT-199 in Biorelevant Media

The determination of ABT-199 solubility was performed by the shake-flask method [17]. Briefly, the apparent solubility of ABT-199 was investigated over 24 h in different biorelevant media (HBSS pH 6.5, FaSSIF, FeSSIF, TIM-fasted, and TIM-fed medium). Fasted state simulated intestinal fluid-V2 powder (FaSSIF-V2) and fed state simulated intestinal fluid (FeSSIF) were prepared based on the recipes described by the vendor. TIM media were obtained during blank runs with the tiny-TIM system in either a fasted or fed state including a high-fat meal. A detailed description of this in vitro tool can be found elsewhere [16,18].

In these experiments, a 50 mg/mL ABT-199 stock solution in DMSO was prepared and 25 µL aliquots were transferred to the respective media to obtain a concentration of 0.25 mg/mL. This allowed for non-sink conditions in all cases and the DMSO concentration not to exceed 0.5% at a given volume for all samples. The sample vials were equilibrated in a shaking incubator over 24 h at 37 °C and 175 rpm. Samples (200 µL) were withdrawn at predefined time points (0.5, 1, 2, 4 and 24 h) and immediately transferred into 1.5 mL Eppendorf tubes for centrifugation (5 min, 37 °C, 21,000 rpm). Then, 100 µL of supernatant were subsequently diluted (1:5) with a methanol:water (70:30, *v*/*v*) mixture, further centrifuged at similar conditions and, finally, transferred into HPLC vials for quantification. The solubility values obtained in the time frame 0.5–4 h were considered as kinetic solubility values, as the system was not in equilibrium. After 24 h, equilibrium conditions were assumed. 

### 2.6. Permeability of ABT-199 in Biorelevant Media

The permeability study for ABT-199 was performed and analyzed similarly to what has already been described in Section 2.4. The only differences were that 1 mg/mL (or 1.15 mM) of ABT-199 stock solution was prepared in DMSO and transferred into different biorelevant media (i.e., HBSS pH 6.5, FaSSIF, FeSSIF, TIM-fasted and TIM-fed). Here, target concentrations of 20, 50 and 100 µg/mL (or 11.5, 23, 57.5 and 115 µM, respectively) were obtained in each medium. The assay was performed for 2 h and 4 h. Due to a fluorescence interference detected from ABT-199, LuY could not be applied in the mixture but was replaced with 10 µM of atenolol as an alternative marker for paracellular integrity. The final concentration of DMSO did not exceed 0.5% at a given volume for all samples. To quantify ABT-199 permeation concentrations, a minimum of eight serial dilutions were prepared by applying a two-fold dilution from 25 µg/mL ABT-199 using HBSS pH 7.5 buffer with 1% BSA. The calibration curve was established using the peak area ratios between the analyte and the internal standard in LC-MS/MS. ABT-199 concentrations were quantified against the calibration. 

### 2.7. Analytics

Quantification of drugs was performed by LC–MS/MS (Acquity UPLC coupled to a Sciex 6500+ or 5500QTRAP, respectively, and operating in multiple reaction monitoring (MRM). A total of 0.025 µM of Carbutamide was used as the internal standard, as mentioned before. Briefly, substances were separated with the BEH C18 or Xbridge C8 columns (30 × 2.1 mm, 1.7 µm; Waters Corp., Milford, MA, USA), using a two-step gradient elution with acetonitrile and water both acidified with 0.1% formic acid. The gradient conditions were 0–0.4 min 1–40% acetonitrile and 0.4–1 min 40–95% acetonitrile with a constant flow rate of 0.7 mL/min. The column temperature was set at 50 °C. Samples were kept at 10 °C. The mass spectrometer was operating in positive electrospray ionization (ESI) mode (5500 V, temperature 550 °C; curtain gas 40 psi, GS1 60 psi and GS2 70 psi). The transition methods are described in Appendix A, for each analyte and its internal standard. Data were processed using Sciex Analyst Software 1.7.2.

### 2.8. Data Analysis and Statistics

All experiments were performed with a minimum of three replicates. Permeability results were presented with mean *P_app_* value ± SD. Comparison between data sets was performed for the paired *t*-test, as indicated accordingly. For significant difference, *p* ≤ 0.05 was considered. Robust non-linear regression plots were performed to assess the sigmoidal correlations between *P_app_* and human *f*_a_ using the four-parameter logistic model fit. The top and bottom constraints from the fitted curves were set at 100 and 0, respectively. Goodness of fit (with quantified R-squared values) and outlier identification were performed by using GraphPad Prism (Version 9.1).

## 3. Results

### 3.1. Effect of Mucin on Monolayer Integrity in Presence of Different Biorelevant Media

Figure 2 illustrates the effect of porcine mucin on the cell layer integrity (using LuY permeation) in the presence of different biorelevant media.

Without mucin, FeSSIF, TIM-fasted and TIM-fed media resulted in increasing leakage of LuY in Caco-2, which were approximately 33, 16 and 66-fold higher than for HBSS pH 6.5 after 4 h of treatment, respectively. FaSSIF and HBSS exhibited minimal differences; both had LuY permeation rates below 1% indicative of qualified monolayer integrity in Caco-2 assays. The different monolayer leakage levels can be explained by the different medium compositions, with the highest leakage exhibited by the TIM-fed medium. 

In parallel, 50 and 100 mg/mL of mucin were applied on top of the Caco-2 monolayer to compare the LuY results in different physiological media. With 50 mg/mL of mucin, FaSSIF and FeSSIF exhibited similar results to HBSS pH 6.5. Moreover, all three media types did not damage the cell layer, as the LuY permeation was below 1%. However, TIM-fasted and -fed media led to an increase in the LuY permeation of 1.2% and 14%, respectively, after 4 h. In comparison to the results with HBSS pH 6.5, these values were two-fold and 27-fold higher. For 100 mg/mL of mucin, the monolayer integrity could be effectively maintained upon exposure to all biorelevant media over 4 h. The LuY permeation values from all four biorelevant media were in the range of 0.25–0.6%, and similar to the HBSS control. Overall, 100 mg/mL of mucin provided sufficient protection to the Caco-2 cells for all media types. Thus, it was used for the following drug permeability studies.

### 3.2. Effect of Mucin on Drugs with High and Low Permeability

Before using biorelevant media as donor media, the physical interaction with mucin was evaluated for 15 different reference drugs (BCS Class I to IV) in Caco-2 using HBSS at pH 6.5 (apical)/pH 7.4 (basolateral). The results are shown in Figure 3a, where the presence of mucin led to a significant decrease in *P_app_* values for both the high-permeability drugs (caffeine, metoprolol and propranolol) and the low-permeability drugs (carvedilol, indinavir, ranitidine, sulfasalazine and atenolol). In addition, the fold change calculated by dividing the mucin-absent *P_app_* with the mucin-present *P_app_* was used to visualize the mucin interaction of the drugs. As shown in Figure 3b, a significant decrease was observed for propranolol and carvedilol after the mucin was applied, where the values for mucin-protected monolayers were 99, 756-fold lower, respectively. In comparison, caffeine, theophylline, indomethacin, metoprolol, indinavir, ranitidine, sulfasalazine, atenolol and cromolyn were approximately in a range of a one to three-fold decrease in the *P_app_* values following the addition of mucin. Piroxicam, naproxen, furosemide and doxorubicin exhibited no change with or without mucin, which suggests the influence of mucin can be considered negligible. 

### 3.3. Permeability of Reference Drugs in TIM Media 

As a follow-up, we correlated the permeability of the reference drugs with dependence of the biorelevant media applied apically to Caco-2 monolayers in presence of 100 mg/mL of mucin with published human intestinal drug absorption (*f*_a_) (Figure 4). Despite a strong sigmoidal correlation with all biorelevant media types, carvedilol and propranolol were consistently identified as outliers. Due to the fact that their Caco-2 *P_app_* values were strongly affected by the presence of mucin (Figure 3b), both drugs were excluded from that correlation. 

To explore the relationship between media system performances for a particular prandial condition (fasted or fed), linear correlations were carried out for all 15 reference drugs to further compare the predictability of SIF with that of TIM-media (Figure 5). A good correlation was observed for log *P_app_* determined in FaSSIF and TIM-fasted, but a decreased correlation was observed between log *P_app_* in FeSSIF and TIM-fed media. FeSSIF and TIM-fed differed primarily for furosemide and naproxen, where permeability was clearly underestimated. 

### 3.4. Solubility of ABT-199 in Biorelevant Media

The solubility of ABT-199 was determined in five different media over a period of 24 h by using the shake-flask method (Figure 6). Highest ABT-199 concentrations were achieved in the TIM-fasted and TIM-fed media. Interestingly, the concentrations in the TIM media remained largely unchanged over 24 h, and drug precipitation was not observed to greatly influence the solubilization. The initial increase in the ABT-199 solubility in the TIM-fasted medium observed during the first two hours of the test could be attributed to a short-termed supersaturation, which is associated with the re-dissolution of amorphous material. The decrease in the ABT-199 concentration suggests a slight precipitation until the equilibrium solubility is reached after 24 h. In the SIF-based media, a much lower solubilization capacity for ABT-199 was measured. These values were consistent with the poor aqueous solubility described in the literature [19]. Here, a declining trend in the solubility was observed in the FaSSIF, suggesting a continuous precipitation of the drug over time. Due to the absence of surfactants, the ABT-199 solubility in HBSS was lowest and limited to a range of 1 to 8 µg/mL over 24 h. The differences in solubility highlight the differences in medium compositions, as described earlier. Specifically, TIM-media are composed of highly biorelevant components, including porcine bile salts, pancreatin, digestive enzymes and digestion products (in the case of the fed state). These induced a higher ABT-199 solubilization capacity than less complex aqueous media, such as SIF or HBSS.

### 3.5. Permeability of ABT-199 in Biorelevant Media

Based on the solubility results in Figure 6, different ABT-199 concentrations (20, 50 and 100 µg/mL) were applied apically in different biorelevant media to Caco-2 monolayers, which were protected with 100 mg/mL of mucin (Figure 7).

The selected target concentrations in the donor (apical) were below the saturation solubility of ABT-199 in both TIM media. However, the lowest concentration (20 µg/mL) corresponded approximately to the kinetic solubility of ABT-199 in the SIF media, and the concentration of 50 µg/mL surpassed the kinetic solubility. 

Since the interference in fluorescence caused by ABT-199 did not allow the use of Lucifer Yellow, atenolol was applied as an alternative marker to monitor the monolayer integrity after treatment with the ABT-199 in biorelevant media. The *P_app_* values of atenolol were similar among the different media, and thus, the monolayers were assumed to be intact during the experiments with ABT-199. 

As can be seen from Figure 7, the use of TIM-fed medium led to relatively higher permeation of ABT-199 in the mucin-protected Caco-2 cells than the use of TIM-fasted medium. Although ABT-199 exhibited concentration-dependent permeation with values above the limit of detection (LOD = 0.009 µg/mL) in LC–MS/MS, the measured concentrations above the lower limit of quantification (LLOQ =0.016 µg/mL) were only achieved when 50 and 100 µg/mL of ABT-199 were applied with TIM-fed media or when 100 µg/mL of ABT-199 were applied with TIM-fasted media. On the contrary, using FaSSIF and FeSSIF as donor media resulted in a lower permeation of ABT-199 across Caco-2 with concentrations below LLOQ. 

Furthermore, 100 ug/mL of ABT-199 in TIM-fed medium achieved 1.6-fold higher permeability compared to the TIM-fasted medium (Figure 7c). Due to the target concentration in both media being below the saturation solubility of ABT-199, the results suggest that permeation was influenced by prandial state-dependent conditions. 

## 4. Discussion

The study of solubility and permeability is expected to provide a deeper understanding of drug absorption, especially in case of drugs, for which solubility and permeability are considered as rate-limiting factors for oral absorption. Here, we present a novel approach to assess solubility and permeability under physiologically relevant conditions based on the media generated by tiny-TIM.

### 4.1. Solubility in Biorelevant Media

Typically, FaSSIF and FeSSIF are considered standard media for solubility and dissolution experiments applied in the biopharmaceutical characterization of novel drugs, as well as in the development of oral drug products [8]. However, they oversimplify the complex in vivo conditions in the human GI tract. For instance, SIF powder is mainly based on sodium taurocholate, whereas, in human intestinal fluids, multiple bile salts can be found [20]. Thus, their solubilization capacity may be limited for certain drugs, and, therefore, the solubility in luminal fluids could be underestimated. In addition, this could also affect the in vitro assessment of drug release from oral formulations. 

A better simulation of the complex composition of GI fluids can be made by advanced dissolution and digestion models based on highly complex media such as tiny-TIM [16,21]. In these models, a pool of different bile salts, pancreatic juice, digestive enzymes and several digestion products are used to mimic digestive processes under simulated GI conditions. 

The ABT-199 solubility data presented in this study revealed that this complex composition of luminal fluids represented an advantage with respect to the assessment of drug solubilization over standard media, such as FaSSIF or FeSSIF. Interestingly, the values obtained here were close to the ABT-199 solubility data reported for porcine intestinal fluids by Henze and colleagues [22]. Obtaining realistic values for solubility of poorly soluble compounds during early drug product development can be regarded as highly beneficial for the selection of the formulation approach, as well as for pharmacokinetic (PK) prediction based on physiologically based pharmacokinetic (PBPK) models.

### 4.2. Effect of Biorelevant Media on Cell Layer Integrity and Protection by Mucin

For combined solubility/permeability experiments with biological membranes, an optimization of the composition of simulated intestinal fluids would come at the cost of compatibility problems with the cells. This issue can be addressed by mucus-protected cell models. For instance, the co-culturing of Caco-2 and mucus-secreting HT29-MTX cells has been proposed in a previous work [13]. However, as compared to the Caco-2 monoculture, this approach has several disadvantages, such as higher paracellular transport caused by the presence of goblet cells [23], impracticality of the co-culturing maintenance for screening assays, inadequate mucus protection, as well as underprediction of the in vitro to in vivo permeability correlation [14]. Another approach is the application of artificial mucus on top of the Caco-2 cells. Although cell-protective mucus has been shown to stabilize cells against certain formulations [12], the experimental outcomes may be affected largely by the presentation of the mucus (i.e., mucus density, mucus type, cell compatibility). To address this question, Boegh and colleagues have carried out studies with biosimilar mucus [24]. Of the different types of mucus tested, one promising approach was to use porcine mucin as a substitute for native mucus. This approach was also previously successfully applied in drug absorption studies with human fasted intestinal fluid [15]. 

In this work, we utilized cell-protective porcine mucin (type III) to enhance Caco-2 compatibility with biorelevant media. Along with FaSSIF and FeSSIF, media obtained from tiny-TIM in either a fasted or fed state were applied on the apical side. These were investigated as alternatives to standard biorelevant media already in previous studies, where cell compatibility was tested by paracellular transport of certain integrity markers. In these experiments, compatibility was found neither for Caco-2 cells, nor for the co-culture with HT29-MTX [25]. 

However, with the application of porcine mucin, the Caco-2 monolayer was protected for over 4 h of exposure from the detrimental effects of all media, including the TIM-media. As shown by the results for the integrity marker LuY, mucin was able to maintain optimal cell integrity and compatibility. This was in stark contrast to the leaky monolayers observed in absence of mucin for FeSSIF as well as the TIM-fasted and TIM-fed media. Due to its lower bile salt concentration, FaSSIF was generally well tolerated by the cells, which was consistent with the findings of previous publications [15]. By increasing the amount of mucin (0–100 mg/mL), even for TIM-fed medium, the LuY leakage could be reduced from ~19% to values below 1%, similar to the HBSS untreated control. Thus, sufficient integrity was achieved at a mucin concentration of 100 mg/mL even for media that contained a high amount of bile salts, pancreatic juice and digestive enzymes.

### 4.3. Mucin Effect on Drug Permeability

The potential for mucin–drug interactions was further investigated by using 15 reference drugs of different BCS classes. It was shown that the addition of mucin caused a drastic reduction in permeability in the cases of propranolol and carvedilol. Here, the calculated fold change (=*P_app_*, _mucin-free_/*P_app_*_, mucin_) exceeded the values of 99 and 756, respectively. For the other 13 drugs, only a maximum three-fold change was observed, and such effect was considered acceptable based on our overall analysis.

### 4.4. Predictive Power of the Mucin-Protected Caco-2 Assay Based on TIM Media

The relationship between human *f*_a_ and the Caco-2 permeabilities obtained in different media revealed that SIF and TIM media presented similar predictive abilities for a series of reference drugs in the fasted state. However, a stronger discrepancy was observed for the permeability values obtained in simulated fed-state conditions with a limited number of drugs. This effect was shown through the stronger correlation observed between TIM-fasted and FaSSIF, as well as the weaker correlation for TIM-fed and FeSSIF. Overall, the predictive capacity of the TIM-based media was generally better than that for the SIF-based media.

Interestingly, propranolol and carvedilol were identified as outliers in all correlations with human *f*_a_ in the sigmoidal relationship analysis. With respect to various physiochemical properties (i.e., ionization behavior, charge at physiological pH, lipophilicity, water solubility, etc.), it is evident that propranolol and carvedilol are both basic molecules with pK_a_ values at around nine. They are therefore positively charged in the small intestinal environment and show a certain degree of lipophilicity (logD_pH 7.4_ ≈ 1.5). This combination of properties may have resulted in specific drug–mucin interactions, which could alter drug diffusion and affinity for these non-glycosylated regions. Moreover, negatively charged functional groups of mucin may also have the propensity to trap positively charged drug molecules [26]. However, it cannot be precluded that other factors may also contribute to pronounced mucin interactions seen for propranolol and carvedilol, since other drugs with similar properties (logP > 1 and positive charge) were not impacted by mucin in terms of their correlation to human *f*_a_. 

The effect of the unstirred water layer (UWL) may be another rate-limiting factor that could impact the values of apparent drug permeability in Caco-2 (with or without mucin), given that the agitation of the experimental solution was not performed during our drug transport study. Therefore, it may be important to test the agitation influence on UWL for mucin–drug interactions and explore the mitigation of mucin and non-mucin discrepancy in cell permeability studies. 

The strong predictive abilities observed for TIM-fasted and TIM-fed media suggested a translational application of accessing prandial state-dependent drug absorption. This could be an important aspect in better understanding the mechanisms leading to food effects on oral drug bioavailability. 

### 4.5. Case Study: ABT-199

Venetoclax (ABT-199) was selected as a poorly permeable and poorly soluble drug probe for a case study. Therefore, ABT-199 represented an ideal candidate to measure permeability in the mucin-protected Caco-2 cells combined with TIM-media. In our study, the solubility of amorphous ABT-199 in TIM media was clearly higher than in SIF-based media. Solubility values in fasted and fed media were comparable, consistent with similar observations made for porcine intestinal fluids [27]. The abundance and complexity of solubilizing agents, i.e., bile salts, phospholipids, as well as protein and lipid digestion products, may be of major importance for the higher concentrations of dissolved ABT-199 in TIM-based media and real luminal fluids [28]. 

Drug concentrations below the saturation solubility were targeted in the ABT-199 permeability study using TIM media to independently assess the effect of medium composition on permeation under similar experimental conditions. A linear, but concentration-dependent, permeation was observed for ABT-199. Higher permeation rates were also observed in the TIM-fed medium than in the TIM-fasted medium, although the monolayer integrity was consistently maintained in all concentrations. The 1.6-fold increase in fasted to fed permeability suggested that the permeation of ABT-199 was dependent on the prandial conditions. With this in mind, and considering that the target ABT-199 concentrations were below saturation solubility in both fasted and fed conditions, the present approach may potentially discriminate according to the influence of the media composition and the species formed in solution upon permeation of ABT-199. 

As a low permeability and low solubility bRo5 drug and with mucin effects, the limit of quantification for ABT-199 was restricted. Therefore, only a few samples could be measured that were above the LOQ in TIM-fed and TIM-fasted media at the highest mucin concentration of 100 mg/mL with an extended incubation time from 2 h (data not shown) to 4 h. In contrast, FaSSIF and FeSSIF were not suitable for achieving meaningful results for ABT-199 in mucin–Caco-2 assays, as the obtained values were either close to the LOD or below the LLOQ. This aspect highlighted the importance of using advanced biorelevant media for permeability studies, as they assist in overcoming the solubility limitations of standard dissolution media for bRo5 drugs. The use of TIM media with mucin-protected Caco-2 cells represents a promising model to better capture drug absorption under physiologically relevant conditions. Moreover, this assay also has the potential to improve PK prediction by providing more realistic estimates of solubility and permeability for PBPK models, as well as to better describe the mechanisms leading to food effects on oral bioavailability.

Additionally, simultaneous investigation of the interplay established between all intraluminal processes and oral absorption could be understood by combining TIM experimental runs with a subsequent transfer of intraluminal samples to the mucin-protected Caco-2 model. In this manner, additional key aspects related to the formulation and physiology of the gastrointestinal tract under fasted or fed conditions could be covered in order to improve the predictability of the combined dissolution/permeation model, as well as to improve the mechanistic understanding and identify critical attributes of a formulation’s performance. 

## 5. Conclusions

The current study highlights the importance of using biorelevant media for the characterization of solubility and permeability of drugs, which are both poorly water soluble and poorly permeable (BCS class IV). As compared to standard media such as HBSS, FaSSIF or FeSSIF, the use of media generated in the tiny-TIM system led to solubility values that were close to the previously published data measured in porcine intestinal fluids. Due to the incompatibility of these media with Caco-2 cells, protection by mucin was necessary and evaluated at different concentrations. The broad validation with 15 reference drugs, as well as the case study performed with ABT-199, revealed that the mucin-protected Caco-2 model represents a promising model for optimal permeation assessment. This study facilitates a more realistic assessment of drug absorption under simulated fasted- and fed-state conditions early in clinical development. 

## Figures and Tables

**Figure 1 pharmaceutics-14-00699-f001:**
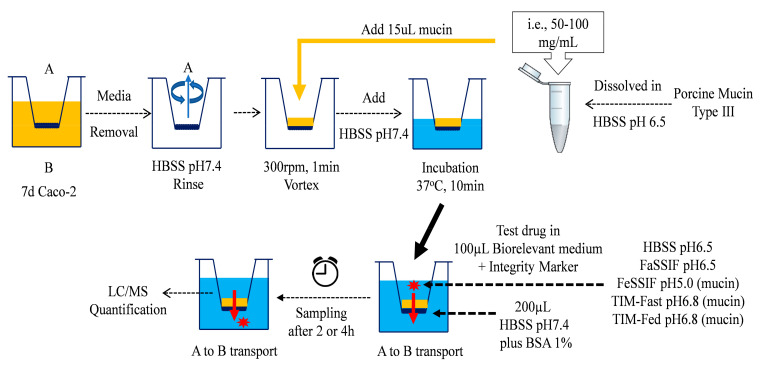
Applications of fasted- and fed-state biorelevant media as the apical (A) media systems in mucin-protected Caco-2 model for the assessment of drug permeability from the basolateral (B) compartment.

**Figure 2 pharmaceutics-14-00699-f002:**
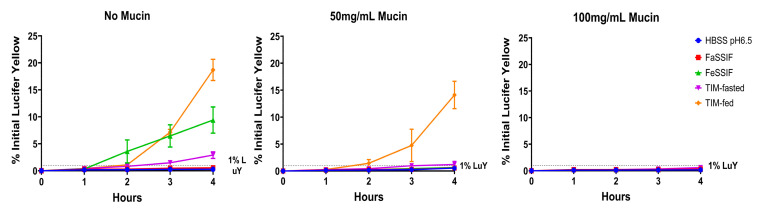
Porcine type III mucin maintains Caco-2 monolayer integrity in concentration dependence during the applications of fasted and fed state SIFs and TIM media. The % LuY permeation was measured over 4 h following the treatment of HBSS pH 6.5, FaSSIF, FeSSIF, TIM-fasted and TIM-fed media in the Caco-2 together with 0, 50 and 100 mg/mL of mucin.

**Figure 3 pharmaceutics-14-00699-f003:**
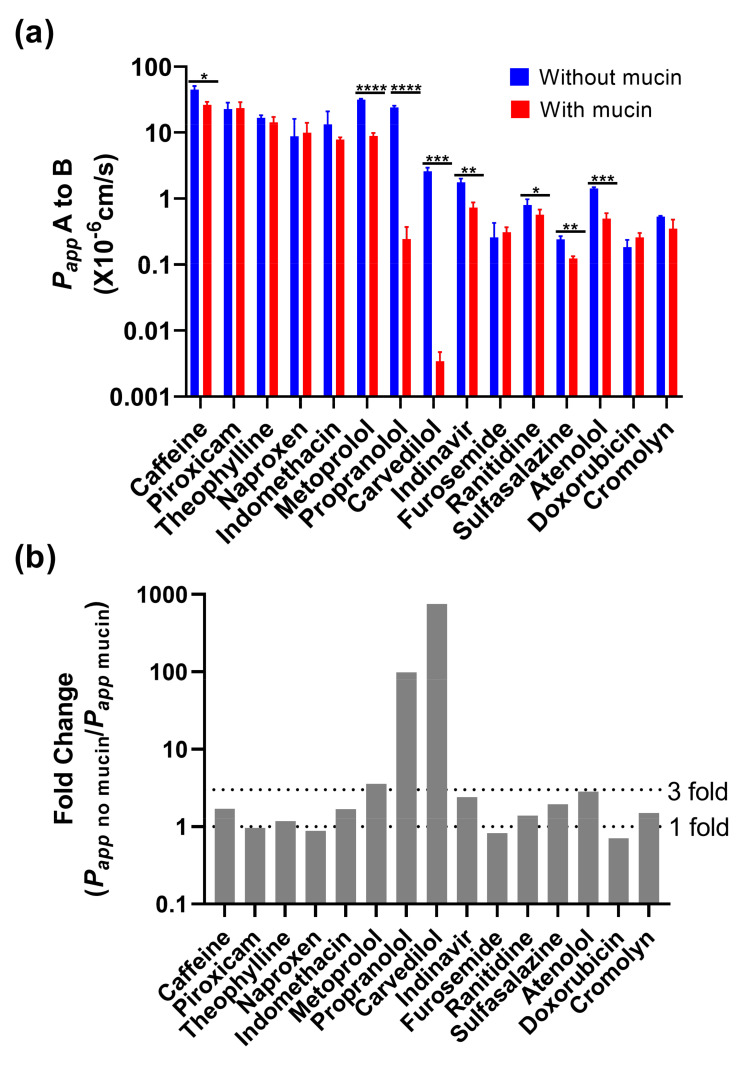
Permeability of 15 reference drugs in the presence or absence of 100 mg/mL of mucin in the Caco-2 permeability assay. (**a**) The drugs were measured in HBSS pH 6.5 with absence and presence of mucin; (**b**) Based on these results, the fold change was calculated by dividing the mucin-free *P_app_* value with the mucin-dependent *P_app_* value. Statistical result was calculated with a paired *t*-test, with significance defined at *p* ≤ 0.05 (*), *p* ≤ 0.01 (**), *p* ≤ 0.001 (***), *p* ≤ 0.0001 (****).

**Figure 4 pharmaceutics-14-00699-f004:**
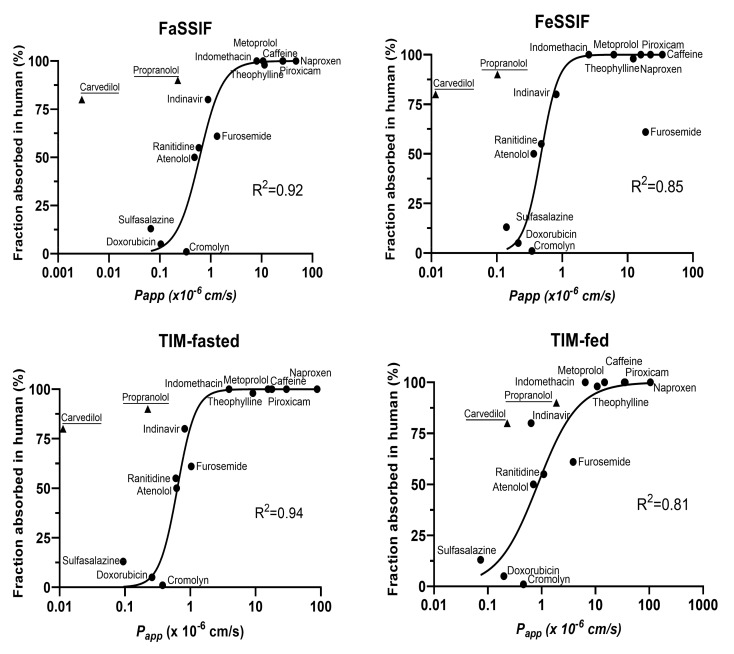
Correlation between Caco-2 *P_app_* measured in different biorelevant media human *f*_a_. The sigmoidal correlations with human *f*_a_ were performed using Caco-2 *P_app_* (propranolol and carvedilol excluded) measured with FaSSIF, FeSSIF, TIM-fasted and TIM-fed.

**Figure 5 pharmaceutics-14-00699-f005:**
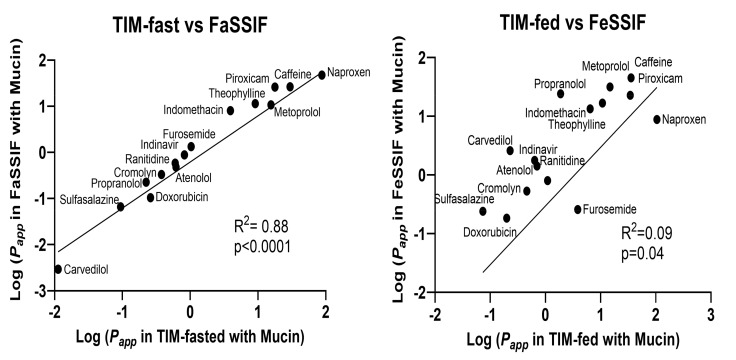
The linear correlations based on Caco-2 *P_app_* values obtained in TIM-fasted and FaSSIF media or TIM-fed and FeSSIF media.

**Figure 6 pharmaceutics-14-00699-f006:**
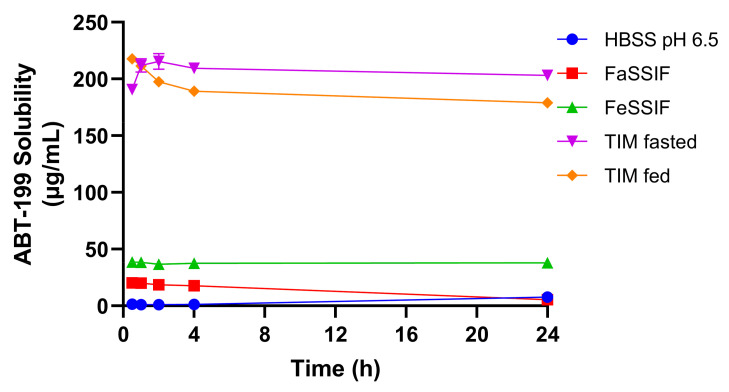
ABT-199 solubility in different biorelevant media over 24 h incubation.

**Figure 7 pharmaceutics-14-00699-f007:**
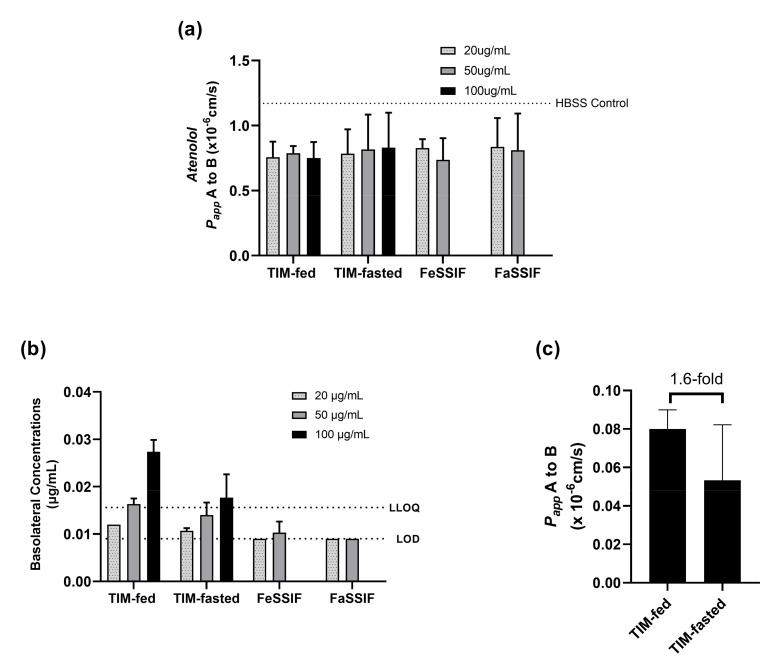
Permeation of ABT-199 across 100 mg/mL of mucin-protected Caco-2 for 4 h in different donor medium systems. (**a**) paracellular permeability (monolayer integrity) of the monolayers was monitored with atenolol for different fasted and fed media (compared to the atenolol permeability in HBSS buffer (dotted line)); (**b**) ABT-199 transported with different donor concentrations; (**c**) permeability of ABT-199 at 100 mg/mL was measured with TIM-fed and fasted media.

**Table 1 pharmaceutics-14-00699-t001:** Physicochemical properties of 15 selected reference drugs.

Drug	MW (g/mol)	BCS	Ionization Behavior ^a^	pKa ^a^	Physiological Charge ^b^	Log P ^b^	Water Solubility (mg/mL) ^b^	Transport Pathway ^a^	Human *f*_a_ ^a^
Caffeine	194	I	Neutral	NA	0	−0.55	11	Transcellular	100
Piroxicam	331	II	Zwitterion	4.0	−1	0.60	0.143	Transcellular	100
Theophylline	180	I	Acid	7.8	0	−0.77	22.9	Transcellular	98
Naproxen	238	II	Acid	4.2	−1	2.99	0.05	Transcellular	100
Indomethacin	357	II	Acid	3.8	−1	3.53	0.002	Transcellular	100
Metoprolol	267	I	Base	9.6	1	1.76	0.402	Transcellular	100
Propranolol	259	I	Base	9.5	1	2.58	0.079	Transcellular	90
Carvedilol	406	II	Base	7.8	1	3.42	0.004	Transcellular	80
Indinavir	712	IV	Base	3.0/7.2	1	2.81	0.048	Transcellular	80
Furosemide	330	IV	Acid	4.2/9.8	−1	1.75	0.118	Paracellular	61
Ranitidine	314	III	Base	8.1	1	0.99	0.08	Paracellular	55
Sulfasalazine	398	IV	Acid	2.4/9.7	−2	3.94	0.046	Transcellular	13
Atenolol	266	III	Base	9.6	1	0.43	0.429	Paracellular	50
Doxorubicin	543	III	Base	7.3	1	0.53	1.18	Paracellular	5
Cromolyn	468	III	Acid	2.0	−2	1.48	0.036	Paracellular	1

^a^ Literature data by Wuyts et al. [15], ^b^ data acquired from DrugBank online database (www.drugbank.com, accessed on 1 July 2021).

## Data Availability

The raw/processed data required to reproduce these findings cannot be shared at this time due to legal or ethical reasons.

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
