# Peer review of "Mucin-Protected Caco-2 Assay to Study Drug Permeation in the Presence of Complex Biorelevant Media"

_pharmaceutics, 2022, doi:10.3390/pharmaceutics14040699_

Round 1

Reviewer 1 Report

The subject of the study falls within the scope of the “Pharmaceutics” since it describes the preparation and evaluation of an advanced mucin protected-Caco-2 assay methodology to allow the assessment of drug permeation in complex biorelevant media.

This is a comprehensive study that developed an optimized, reliable, and robust method to simulate and evaluate in vitro the intestinal drug permeation, allowing the investigation of the effect of complex biorelevant media mimicking both fasted and fed state conditions.

The study design is very meticulous, and the authors have done a great job describing the development, validation and evaluation of the proposed mucin protected-Caco-2 assay methodology.

Some minor comments are listed below:

  1. In page 8, renumber Figure 1 to Figure 3.
  2. In page 11, Figure 6 is not referred in the text.
  3. In page 11, regarding “Permeability of ABT-199 in biorelevant media” are there any control data of ABT-199 permeability from non-mucin protected Caco-2 cells using the standard HBSS pH 6.5 medium?

Reviewer 2 Report

Recommendation: minor revisions needed.

1. This topic is interesting, and the results could support the conclusion. This article discovers the Mucin Protected Caco-2 assay to study drug permeation in the presence of complex biorelevant media and better described the mechanisms that lead to the effects of food on oral bioavailability. However, oral absorption of drugs is a complex process, so the mechanism of bioavailability needs further experimental research and supplementary interpretation of related experiments. As such, I recommend that this paper may be published in Pharmaceutics after minor corrections. 

2. In Figure 6, we observed that the ABT-199 solubility curve displayed by TIM-fasted showed a phenomenon of first rising and then falling. Please explain this phenomenon appropriately.

3. Due to the complexity of oral absorption of drugs, in the solubility test, different pH conditions should be selected for the experiment.

4. When the abbreviation appears for the first time in the text, please use the full name. Such as PK, PBPK, etc.

5. In order to better review the article, please add the line number of the article when submitting the manuscript.

Reviewer 3 Report

Dear Authors:

I read your interesting manuscript and felt that the results were not shown properly.  I am afraid that it might be hard to follow.  

  1. Please check English entirely once again.  I found some grammatic errors.  For example, line 1 of page 13.  "...cells has been.."
  2. Correct the font of eq.1 at page 4.  
  3. Please revise the discussion part.  It looks a bit diffuse and the main point went out of focus, so I felt.  Hopefully, you should have focused on your explanation, ideas, deduction, reasoning about why Mucin-protected Caco-2 assay could improve permeation.  Would you mind using schematic figures etc. to arrange this part? 
  4. Would you mind making clear your experimental results, summarizing them shortly and bulletizing them?  

Round 2

Reviewer 3 Report

Dear authors:

I confirmed that you sincerely revsed your original manuscript to improve the quality.  Hopefully, your published manusript will attract readers much and bring them useful information in the filed.